# Antibacterial Potential of *Bacillus amyloliquefaciens* GJ1 against Citrus Huanglongbing

**DOI:** 10.3390/plants10020261

**Published:** 2021-01-29

**Authors:** Jing Nan, Shaoran Zhang, Ling Jiang

**Affiliations:** 1College of Horticulture and Forestry, Ministry of Education Key Laboratory of Plant Biology, Huazhong Agricultural University, Wuhan 430070, China; nanjing@webmail.hzau.edu.cn; 2State Key Laboratory of Agricultural Microbiology, Huazhong Agricultural University, Wuhan 430070, China; zsr@mail.hzau.edu.cn

**Keywords:** huanglongbing, *Bacillus amyloliquefaciens* GJ1, biocontrol, induced systemic resistance

## Abstract

Citrus huanglongbing (HLB) is a destructive disease caused by *Candidatus* Liberibacter species and is a serious global concern for the citrus industry. To date, there is no established strategy for control of this disease. Previously, *Bacillus amyloliquefaciens* GJ1 was screened as the biocontrol agent against HLB. In this study, two-year-old citrus infected by *Ca.* L. asiaticus were treated with *B. amyloliquefaciens* GJ1 solution via root irrigation. In these plants, after seven irrigation treatments, the results indicated that the photosynthetic parameters, chlorophyll content, resistance-associated enzyme content and the expression of defense-related genes were significantly higher than for the plants treated with the same volume water. The content of starch and soluble sugar were significantly lower, compared to the control treatment. The parallel reaction monitoring (PRM) results revealed that treatment with *B. amyloliquefaciens* GJ1 solution, the expression levels of 3 proteins with photosynthetic function were upregulated in citrus leaves. The accumulation of reactive oxygen species (ROS) in citrus leaves treated with *B. amyloliquefaciens* GJ1 flag22 was significantly higher than untreated plants and induced the defense-related gene expression in citrus. Finally, surfactin was identified from the fermentation broth of *B. amyloliquefaciens* GJ1 by high-performance liquid chromatography. These results indicate that *B. amyloliquefaciens* GJ1 may improve the immunity of citrus by increasing the photosynthesis and enhancing the expression of the resistance-related genes.

## 1. Introduction

Huanglongbing (HLB, also known as citrus greening) is the most devastating disease of citrus and has caused substantial crop losses. HLB is associated with three species of Gram-negative, α-proteobacteria, *Candidatus* Liberibacter spp., namely, “*Ca*. L. asiaticus”, “*Ca*. L. africanus”, and “*Ca*. L. americanus” [1]. The pathogen is transmitted by the citrus psyllid and resides inside the phloem elements. Among them, “*Ca*. L. asiaticus” is the most widespread [2]. Typical symptoms of HLB disease include yellow shoots; blotchy mottling leaves; upright, hardened, and small leaves; leaves showing zinc deficiency and corky vein; twig dieback; stunted growth; and tree decline [3]. All varieties of cultivated citrus species are susceptible to HLB [4,5,6]. HLB is rapidly spreading and is having a devastating impact on citrus production worldwide. The effects of HLB have been confirmed in 51 of the 140 citrus producing countries [7]. Infected citrus trees have a dramatically shortened profitable life and their yield is significantly reduced [8]. The disease has caused the loss of approximately 100,000 acres of citrus since HLB was first discovered in Florida in 2005 [9].

Various control measures have been applied to slow the progress of HLB and help maintain the yield under field conditions. A primary approach for HLB management is the control of the psyllid vectors. Controlling the vectors slows down the spread of disease and lessening the severity of HLB [10]. Enhanced foliar nutritional programs and foliar application of inorganic phosphorus solution improved fruit production [11]. Application of antibiotics including penicillin G, oxytetracycline, and streptomycin decreases the population density of the pathogen [12,13,14]. Thermotherapy suppresses phytopathogen titer and reduces the impact of HLB. However, the effectiveness of these treatments is often inconsistent or controversial under field conditions [15,16]. Thus, efficient and sustainable approaches to control HLB are urgently needed.

Biological control efforts, using microorganisms as agents for combatting plant pathogens. Plant growth-promoting rhizobacteria (PGPR) are a group of bacteria that generally colonize in the rhizospheric zone of the plant tissue [17]. Use of PGPR offers an ecological method to manage pathogens in agriculture [18]. PGPR facilitate enhancing plant growth through various activities, such as mobilization of soil nutrients [19], synthesis of phytohormones [20], and secretion of plant growth regulators [21]. PGPR also produce antimicrobial compounds, such as bacteriocin, which can target pathogens and reduce plant diseases [22]. PGPR can trigger systemic resistance as a result of colonization of plant roots. Induction of resistance is achieved by the direct activation of defenses, and also achieved by priming [23]. The combination of direct induction and priming results in enhanced plant defense capacity.

Our laboratory isolated *B. amyloliquefaciens* GJ1 from the leaves of citrus that were fighting against huanglongbing [24]. In this study, Ca. L. asiaticus-infected citrus were used for treatment with *B. amyloliquefaciens* GJ1 and had three objectives. The first objective was to find the effect of *B. amyloliquefaciens* GJ1 treatment on photosynthesis, because phloem blockage is a major reason for HLB disease symptom development [25]. The second was to determine the changes of defense-related genes in the transcript levels after treatment with *B. amyloliquefaciens* GJ1. The final objective was to determine whether treatment with *B. amyloliquefaciens* GJ1 flag22 causes changes in the transcript levels of defense-related genes, specifically those of *WRKY*22 and *GST*1.

## 2. Results

### 2.1. B. amyloliquefaciens GJ1 Increased Photosynthesis

After *B. amyloliquefaciens* GJ1 solution was applied seven times, some indices (photosynthesis activity, starch, and soluble sugar content) of plant leaves were significantly affected. The photosynthesis activity and the total chlorophyll content increased by 13.75% and 22.27%, respectively, compared with those of the control treatment with water (Figure 1a,b). After seven treatments with *B. amyloliquefaciens* GJ1 solution, the starch and soluble sugar contents of the mature leaves were decreased significantly (*p* < 0.05) compared with those of the control. Starch showed 23.43% reduction (Figure 1c). Fructose, maltose, and glucose showed 55.81%, 33.03%, and 59.05% reductions, respectively (Figure 1d–f).

### 2.2. Parallel Reaction Monitoring Quantification of Three Candidate Proteins

In our previous work, the photosynthesis pathway of citrus was significantly upregulated after treatment with *B. amyloliquefaciens* GJ1. *Cs2g07330*, *Cs5g34450*, *Cs5g31180*, *Cs6g12390*, *Cs3g06180*, *Cs5g18620*, and *orange1.1t03504* were upregulated by 1.39-, 1.67-, 1.44-, 1.81-, 1.48-, 1.39-, and 1.88-fold, respectively [24]. In this study, the expression levels of three candidate proteins with photosynthesis functions were chosen for quantification by PRM to verify the transcriptome results. The PRM results showed in Table 1, and the abundance of PsbQ, PsaE, and PsaD changed in a consistent way with the transcriptome results (Figure 2).

### 2.3. B. amyloliquefaciens GJ1 Treatment Increases the Content and the Transcript Level of Malic Enzyme and Transketolase

Malic enzymes are involved in many metabolic pathways. In plants, they can provide carbon dioxide for photosynthesis [26]. Plant transketase is involved in the photosynthetic carbon cycle, which plays an important role in carbon fixation [27]. Thus, the changes in malic enzyme and transketolase were determined after the treatment with *B. amyloliquefaciens* GJ1. The malic enzyme content in mature leaves and fibrous roots increased from 7.76 mIU/g and 7.86mIU/g to 11.08 mIU/g and 9.01 mIU/g, respectively (Figure 3a). The content of transketolase in mature leaves and fibrous roots increased from 11.21 mIU/g and 7.40 mIU/g to 28.63 mIU/g and 10.84 mIU/g, respectively (Figure 3b).

The mature leaves and fibrous roots of the control group and the group treated with *B. amyloliquefaciens* GJ1 were collected, and the gene expression of malic enzyme and transketolase was analyzed using RT-qPCR (QuantStudio 6, Waltham, USA). The expression of malase enzyme in mature leaves and fibrous roots increased by 4.58 and 2.13 times, respectively (Figure 3c). The expression of transketolase in mature leaves and fibrous roots increased by 4.88 and 3.59 times, respectively (Figure 3d).

### 2.4. B. amyloliquefaciens GJ1 Treatment Increases the Transcript Level of Defense-Related Genes

The transcript levels of selected defense-related genes were studied in infected citrus leaves after seven irrigation treatments with *B. amyloliquefaciens* GJ1. *WRKY22* and *GST1*, the microbe-associated molecular pattern (MAMP)-triggered immunity (MTI) marker genes, have previously been shown to be responsive to MAMP treatment in citrus [28,29], and some other genes (*GST1*, *WRKY24*, *WRKY33*, *nho1*, and *HSP90*), and were selected for expression analysis. Quantitative RT-qPCR was performed to determine the transcript abundance of these genes. The expression of these genes significantly increased at 6 h after treatment with *B. amyloliquefaciens* GJ1. The transcript levels of *WRKY24* were higher (eight-fold) than those of the control treatment, and the transcript levels of *GST1* and *WRKY22* were 4.5- and 7.5- fold, respectively (Figure 4).

### 2.5. B. amyloliquefaciens GJ1-flag22 Triggers a ROS Burst

Flag22 from the highly conserved *N*-terminal domain of flagellin was the best characterized MAMP. The production of reactive oxygen species (ROS) is a common immune response in plants. Using the luminol assay and histochemical detection assay, we found that *B. amyloliquefaciens* GJ1-flag22 treatment induced ROS production (Figure 5). To determine whether *P. syringae* DC3000 flag22 and Ca. L. asiaticus flag22 could induce ROS production, we then tested in the luminol assay. *P. syringae* DC3000 flag22 elicited high levels of ROS production and *Ca.* L. asiaticus flag22 showed an about 10 min delayed and weak ROS burst.

### 2.6. B. amyloliquefaciens GJ1-flag22 Increases the Content of GSHE, Callose, and the Transcript Level of Defense-Related Genes

Plant immune response to biotrophic pathogens can be divided into microbe-associated molecular pattern (MAMP)-triggered immunity (MTI) and effector-triggered immunity (ETI), and the best characterized MAMP is flag22. MTI induces a suite of immune responses, including ROS production, callose deposition, and defense gene expression. Two-week-old seedling grapefruit were treated with *B. amyloliquefaciens* GJ1-flag22 by trunk injection and after 6 h, the leaves were harvested. The content of GSHE and callose were significantly higher than those of the control treatment. The content of GSHE and callose increased from 67.68 ng/g and 648.6 ng/g to 77.78 ng/g and 1055 ng/g, respectively (Figure 6a,b). The content of ROS was significantly higher in treatments with *B. amyloliquefaciens* GJ1-flag22 as compared with those of the control treatment (Figure 6c).

The transcript levels of the selected defense genes were studied in citrus plants after treatment with *B. amyloliquefaciens* GJ1-flag22. The expression levels of these genes were significantly (*p* < 0.05) increased compared with those of the control, although the fold change was lower than *B. amyloliquefaciens* GJ1 treatment except HSP90 (Figure 7).

### 2.7. Production of Antibacterial Peptides

Producing antimicrobial substances is the most important biocontrol mechanism for *Bacillus* against pathogens. *Bacillus velezensis* FZB42 produce surfactin, fengycin, iturin A, macrolactin, difficidin, bacillaene, bacilysin, and bacillibactin, and we use the primers designed from *Bacillus amyloliquefaciens* GA1 to amplify these genes in *B. amyloliquefaciens* GJ1 [30]. PCR amplification was performed using the DNA of *B. amyloliquefaciens* GJ1 as template. *bacA*, *beaS*, *dfnM*, *dfnA*, *dhbA*, *fenA*, *fenE*, *ituF1*, *mlnA*, *mlnI*, *srfAA*, *srfAD*, *ywfG*, *mrsK2*, *mrsK2R2*, and *mrsR2* were identified (Figure 8), and these genes had a high amino acid identity with corresponding genes in *B. amyloliquefaciens* FZB42 (Table 2).

The supernatants of *B. amyloliquefaciens* GJ1 were collected after 36 h of growth and analyzed using HPLC. Surfactin characteristic peaks appeared with the retention time of 11.549 min, 13.274 min, 13.741 min, and 15.511 min. Under the same elution condition, the characteristic peaks of *B. amyloliquefaciens* GJ1 crude extract appeared at 11.501 min, 13.232 min 13.723 min, and 15.706 min, and the retention time was near the standard peak. It is speculated that surfactin homologs may be present in the crude extract (Figure 9).

## 3. Discussion

Citrus infected by “*Candidatus* Liberibacter asiaticus” causes phloem blockage, then the starch and sugar in leaves accumulate. Excessive starch damages chloroplast function and finally photosynthesis is repressed. A previous study found that *B. amyloliquefaciens* GJ1 could reduce the infection rate of HLB [24]. In this study, Ca. L. asiaticus infected citrus was used for treatment with *B. amyloliquefaciens* GJ1, and the contents of starch and soluble sugar in leaves were significantly decreased compared with those of the control treatment. Compared with the control treatment, the content of resistance-associated enzyme and the expression of defense-related genes were significantly higher. These results demonstrate that *B. amyloliquefaciens* GJ1 could improve the photosynthesis ability of citrus plants and enhance the transport from source to reservoir.

### 3.1. Bacillus Species as Potential Biocontrol Agents against Citrus Huanglongbing

Citrus huanglongbing is a major threat to citrus production and chemical control is an efficient method of controlling this disease. Meanwhile, chemical agents have been widely used in the controlling agricultural diseases (Table 3), but application of antibiotics has a negative impact on the environment and side effects in humans. Biological control providing a safe and effective method of preventing plant diseases. Application of microbial antagonists, especially bacillus spp. and yeast, is the method of controlling agricultural diseases with the most promising potential [31,32,33]. *Bacillus* spp. are the most common endophytic bacteria and have remarkable biological function against plant diseases [34]. Many studies have shown that members of *Bacillus*, such as *B. subtilis*, *B. amyloliquefaciens*, *B. pumilus*, and *B. cereus* can be used as biological agents to control citrus diseases.

*B. subtilis*, *B. cereus*, *B. pumilus*, and *B. megaterium* act against chloromycosis by inhibiting the mycelial growth and spore germination of *P. digitatum* [38]. *B. subtilis* endospores and their antifungal metabolites showed strong antagonistic activity [39,40]. *B. subtilis* could control the fruit drop of citrus caused by *C. acutatum* [41]. *B. subtilis* AF 1 was used as biological agent to control soft rot in lemons [42]. *B. subtilis* LE24, *B. amyloliquefaciens* LE109, and *B. tequilensis* PO80 isolated from citrus plants can control the occurrence of citrus canker in lime [43].

*B. amyloliquefaciens* is usually isolated from the rhizosphere and soil on the surface of plants and can be used as a biological control agent. This strategy has been reported many times in citrus. In vitro and in vivo, *B. amyloliquefaciens* has obvious inhibitory effects on citrus pathogens, such as *P. digitatum*, *P. burosum*, *P. italicum*, and *G. citri-aurantii* [44,45,46]. *B. amyloliquefaciens* can affect the conidia production and mycelial structure of the pathogen of lemon leaf blight in citrus seedlings [47]. *Bacillus* w176 can control the production of green mold in citrus fruits [48]. *Bacillus thuringiensis* can inhibit the growth of citrus coccidia hyphae to control citrus black spot disease [49]. In recent years, the use of *Bacillus* in the prevention and control of citrus diseases has attracted great attention. However, no relevant reports exist on the biological control of HLB, which is a devastating citrus disease. Based on the above background, this experiment found that *B. amyloliquefaciens* GJ1 can effectively prevent and control HLB, which provides new ideas and methods for HLB prevention and treatment.

### 3.2. Mechanism of B. amyloliquefaciens GJ1 in Preventing and Controlling HLB

Leaf mottled yellowing is a typical symptom of huanglongbing disease. HLB causes the accumulation of leaf starch, which prevents the transport of photosynthetic products [50,51]. The host-source-reservoir metabolic imbalance caused by pathogens may be the main cause of huanglongbing disease symptoms [52]. In sweet oranges, grapefruits, and lemons, HLB-infected plants have down-regulated photosystemⅠand photosystem Ⅱ genes at the transcription and protein levels [53,54]. Citrus were treated with *B. amyloliquefaciens* GJ1 mainly affects four KEGG pathways, including the antenna protein and photosynthesis pathway [24]. Light and related genes are up-regulated at the transcription and protein levels. Finally, the light and ability of plants are enhanced and the balance in carbohydrate metabolism is improved. Starch accumulation in the leaves will result in feedback inhibition of photosynthesis, and too much starch will damage the thylakoids of the chloroplast. This study found that after treatment with *B. amyloliquefaciens* GJ1, the photosynthesis of the plant was enhanced, and the total chlorophyll content increased. Photosynthesis increased, but the starch content in the leaves decreased, indicating that *B. amyloliquefaciens* GJ1 treatment enhanced the transport of citrus seedlings from the source to the sink, and phloem blockage was relieved. Induction of systemic resistance is a well-known mechanism for inducing plant host defense responses. Plant growth-promoting bacteria induce the expression of target genes to resist pathogen infection [55,56].

*Bacillus* ABS-S14 can stimulate the expression of defense-related genes to prevent and treat citrus green mold [57]. Induction of systemic resistance is the main antagonistic mechanism of *B*. *amyloliquefaciens* to control tomato fusarium wilt [58]. Therefore, we determined the changes in plant resistance gene expression after treatment with *B. amyloliquefaciens* GJ1. *WRKY22* and *GST1* are the marker genes for microbes to stimulate immunity [28,29]. After treatment with *B. amyloliquefaciens* GJ1, the expression of *WRKY22* and *GST1* was significantly up-regulated, indicating the occurrence of plant immunity.

Microbial-related molecular patterns are pattern recognition receptor-binding ligands [59]. Flag22 is a highly conserved functional domain of flagellin. The flag22 of *Pseudomonas aeruginosa* can bind to the pattern recognition receptor FLS2 of *Arabidopsis thaliana* to trigger immunity [60,61,62]. Immune responses triggered by pathogen-related molecular patterns include the production of ROS, stomata closure, corpus callosum accumulation, expression of defense genes, and hormone biosynthesis [62,63]. In this experiment, 1 µM flag22 of *B. amyloliquefaciens* GJ1 was injected into the stem in this experiment, and the glutathionase and callose contents of the treated group were significantly increased. The ROS of the treatment group was higher than that of the control group, and the expression of resistance-related genes also increased significantly. There is an urgent need to find new prevention methods for pest control, because the chemicals used have a negative impact on the environment and human life, and because of the consumer’s demand for sustainable agriculture. *Bacillus* bacteria have remarkable potential to produce highly active lipopeptides that inhibit insects, mites, nematodes, and plant pathogens. The biological activity mainly exists in three *Bacillus* lipopeptide families: surfacin, iturin, and fengycin [64]. These molecules are involved in the formation of biofilms and induce plant systemic resistance [65]. We used HPLC to analyze the crude extract of *B. amyloliquefaciens* GJ1, and found that *B. amyloliquefaciens* GJ1 can produce surfactin lipopeptide antibiotics.

## 4. Materials and Methods

### 4.1. Plant Materials and Bacterial Growth Condition

In our previous work, our laboratory isolated *B. amyloliquefaciens* GJ1 from healthy leaves of *Citrus sinensis* (L.) Osbeck “Newhall” [24]. *B. amyloliquefaciens* GJ1 was grown in nutrient broth medium at 28 °C, at 150 r/min. *B. amyloliquefaciens* GJ1 was maintained as a glycerol stock at −80 °C. *Citrus sinensis* (L.) Osbeck was used as a scion and grafted onto *Citrus tangerine* Tanaka grown in a greenhouse, and two-year-old plants were identified as HLB-carrying plants by qPCR. Each plant was irrigated with 1.5 L of *B. amyloliquefaciens* GJ1 solution (1.26 × 10^9^ CFU/mL) once every 7 days, and this experiment lasted for 7 weeks. In the end, the mature leaves (fourth leaf from top to bottom) of 8 *B. amyloliquefaciens* GJ1-treated and 8 untreated plants were collected for analysis. All these samples were immediately frozen in liquid nitrogen and stored at −80 °C.

### 4.2. Photosynthetic Parameters, Chlorophyll Content, Soluble Sugar, GSHE, and Callose Content Measurements

The net photosynthetic rate (Pn, mmol CO_2_ m^−2^·s^−1^) was measured using an LI-6400 (LICOR, Lincoln, NE, USA) at steady state under light saturation (1200 mmol m^−2^·s^−1^) and 400 ppm CO_2_. One measurement per plant was performed on the third or fourth leaf from the shoot apex. Five plants were measured for each treatment.

The leaf chlorophyll and carotenoid content of the plants were measured based on the method of Wei et al. [66]. Fresh leaf tissue (0.2 g) was homogenized in 10 mL of 80% acetone and kept for 15 min in the dark, and then centrifuged at 10,000 rpm for 15 min. The absorbance of the supernatant was measured at 663, 644, and 470 nm using a spectrophotometer (Shimadzu UV-1800, Shimadzu corporation, Kyoto, Japan). Total chlorophyll and carotenoid concentration were calculated in terms of fresh weight (FW). Four replicates were used per treatment.

The total soluble sugar was measured by gas-liquid chromatography [67].

GSHE and callose were measured by an enzyme-linked immunosorbent assay (ELISA) kit (Jiangsu Meimian Industry Co., Ltd., Wuxi, China) according to the manufacturer’s instructions.

### 4.3. Parallel Reaction Monitoring (PRM) Analysis

Three *B. amyloliquefaciens* GJ1-treated and three untreated plants were used for PRM analysis. After treatment, the mature leaves were harvest and were ground into powder in liquid nitrogen, mixed with 40 mL of TCA/acetone (1:9), and incubated at −20 °C overnight. The mixture was centrifuged at 7000× *g* for 30 min, the supernatant was discarded, and the precipitate was washed twice with acetone. After drying, the pellet was added with 800 μL of lysis buffer (consisting of 4% SDS, 100 mM Tris-HCl, and 1 mM DTT). The samples were sonicated and then centrifuged at 14,000× *g* for 25 min. The supernatant was a protein extract. Protein concentration was determined with the BCA protein assay reagent [68].

Exactly 40 μL of each protein extract was used for digestion. The final concentration of DTT was 100 mM. The sample solutions were heated at 100 °C for 5 min. After the samples were cooled to room temperature and mixed with 200 μL of UA buffer (8 M Urea, 150 mM Tris-HCl, pH 8.0). The samples were centrifuged at 14,000× *g* for 15 min using an ultrafiltration filter and 200 μL of UA buffer was added before centrifuging again and discarding the filtrate. Next, 100 μL of IAA (50 mM IAA in UA) was added to the samples, which were incubated for 30 min at room temperature in the dark, and subsequently centrifuged at 14,000× *g* for 10 min. The filters were washed with 100 μL of UA buffer and centrifuged twice at 14,000× *g* for 10 min. Dissolution buffer (100 μL of 50 mM triethylammonium bicarbonate pH 8.5) was then added to the filter, and the samples were centrifuged twice at 14,000× *g* for 10 min. Then, 40 μL of trypsin buffer (7 μg trypsin in 40 μL dissolution buffer) was added to the protein suspensions, and incubated at 37 °C for 18 h. The filter was transferred to a new tube and centrifuged at 14,000× *g* for 10 min. The last collected filtrate was analyzed using UV light spectral intensity at OD_280_ [69].

### 4.4. Quantitative Real-Time PCR (qRT-PCR) Analysis

The mature leaves were harvested after seven irrigation treatments, flash-cooled in liquid nitrogen, and stored at −80 °C for subsequent qRT-PCR analysis. Total RNA from citrus leaves was extracted using *Trans*Zol Up ReaFgent (TransGen Biotech, Beijing, China). The integrity of the extracted RNA was checked on agarose gel electrophoresis, and its purity and concentration were assessed using an ND-1000 spectrophotometer (NanoDrop Technologies, Wilmington, USA). Complementary DNA (cDNA) was obtained using a HiFiScript cDNA Synthesis kit (CoWin Biosciences, Beijing, China), following the manufacturer’s instructions. The primers designed in Integrated DNA Technologies are listed in Table 4. The *actin* gene was chosen as the constitutively expressed internal control for normalization [70], and the relative gene expression was calculated using 2^−ΔΔCT^ method.

### 4.5. ROS Production Assays

The luminol assay was performed as previously described (Baker and Mock, 2004). *P. syringae* DC3000 flg22 sequence is TRLSSGLKINSAKDDAAGLQIA, *B. amyloliquefaciens* GJ1 flg22 sequence is EKLSSGLRINRAGDDAAGLAIS, and Ca. L. asiaticus flg22 sequence is DRVSSGLRVSDAADNAAYWSIA. All peptides at >95% purity were synthesized commercially (GenScript, Nanjing, China). All peptides were dissolved in water. In brief, leaf disks (5 mm in diameter) were obtained from young leaves by coring, and the leaf pieces were allowed to float on the water in the Petri dishes for incubation overnight. The next day, one leaf piece was carefully added per well, and then 100 μM luminol, 10 μg/mL horseradish peroxidase, and flag22 (1 μM) were added to each well. Luminescence was measured using the victor nivo plate reader. The content of ROS was measured using an ELISA kit (Jiangsu Meimian Industry Co., Ltd., Wuxi, China) according to the manufacturer’s instructions.

### 4.6. Histochemical Staining of ROS

Histochemical staining with 3,3′-diaminobenzidine (DAB) and nitroblue tetrazolium (NBT) was used to examine the accumulation of H_2_O_2_ and O^2−^, respectively. The leaves from two months old seedling grapefruit were obtained, immersed in prepared 0.5 mg/mL of NBT solutions, and incubated in the dark for 2 h with shaking. The cut leaves were dipped immediately into the DAB solution (1 mg/mL), and then incubated at room temperature for 8 h under room light. The leaves were cleared by boiling in 60% ethanol for 10 min and stored in 75% ethanol until cleared.

## 5. Conclusions

In this study, Ca. L. asiaticus infected citrus were treated with *B. amyloliquefaciens* GJ1 via root irrigation, the photosynthetic parameters, the abundance of three protein in the photosynthesis pathway and the expression of defense-related genes were significantly higher than the control group. *B. amyloliquefaciens* GJ1 flag22 triggered a ROS burst and induced defense gene expression. These results indicated that *B. amyloliquefaciens* GJ1 increasing the photosynthesis and enhancing the expression of the resistance-related genes, then showed antagonistic activity against citrus huanglongbing. Our study also suggested that *B. amyloliquefaciens* GJ1 flag22 is characterized as a potential inducer of defense responses.

## Figures and Tables

**Figure 1 plants-10-00261-f001:**
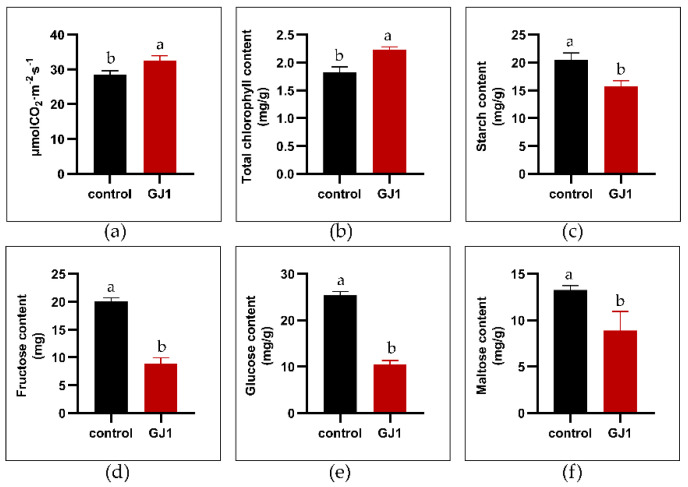
Effect of *Bacillus amyloliquefaciens* GJ1 treatment on some indices of citrus leaves. After seven treatments with *B. amyloliquefaciens* GJ1 solution, the leaves were harvested for analysis. (**a**) The net photosynthetic rate, (**b**) chlorophyll content, (**c**) starch content, (**d**) fructose content, (**e**) maltose content, and (**f**) glucose content. Bars with different letters (a, b) indicate significant differences (*p* < 0.05) between different treatments.

**Figure 2 plants-10-00261-f002:**
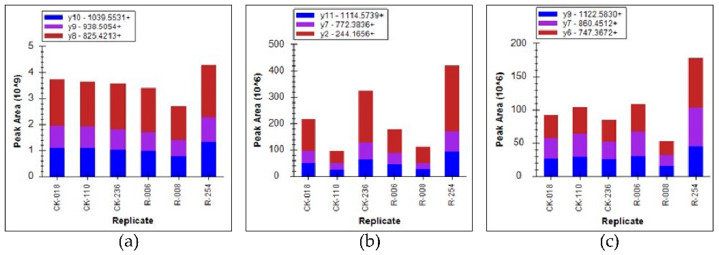
Protein validation of *Cs2g07330* (**a**), *Cs5g34450* (**b**), and *Cs5g31180* (**c**). The bar with different colors represent different peptides.

**Figure 3 plants-10-00261-f003:**
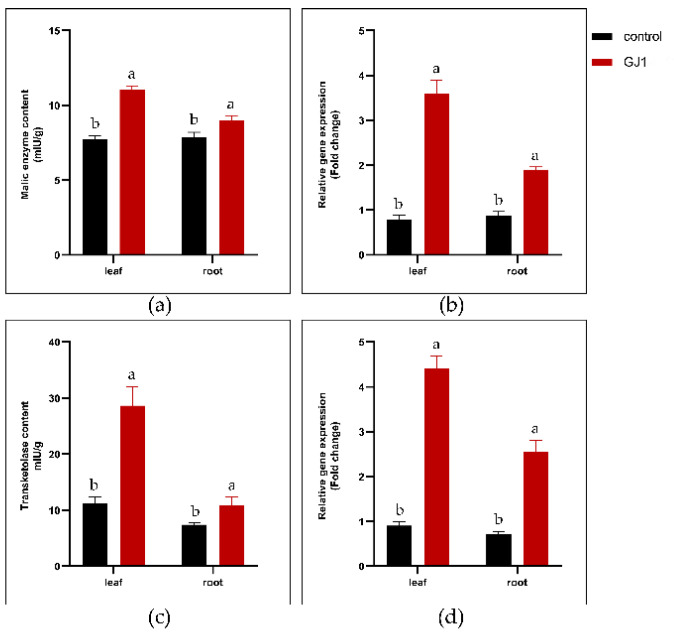
Responses of malic enzyme and transketolase to *B. amyloliquefaciens* GJ1 treatment. Effect of *B. amyloliquefaciens* GJ 1 treatment on the malic enzyme (**a**) and transketolase (**b**) content; effect of *B. amyloliquefaciens* GJ 1 treatment on the transcript level of malic enzyme (**c**) and transketolase (**d**). Bars with different letters (a, b) indicate significant differences (*p* < 0.05) between different treatments.

**Figure 4 plants-10-00261-f004:**
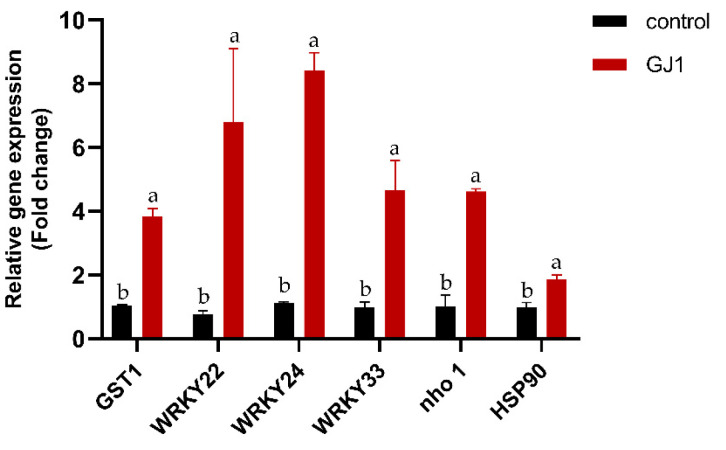
*B. amyloliquefaciens* GJ1 treatment induced an increase in the transcript level of defense-related genes. After treatment, the leaves were harvested and used for RNA extraction and real-time qPCR. Bars with different letters (a, b) indicate significant differences (*p* < 0.05) between different treatments.

**Figure 5 plants-10-00261-f005:**
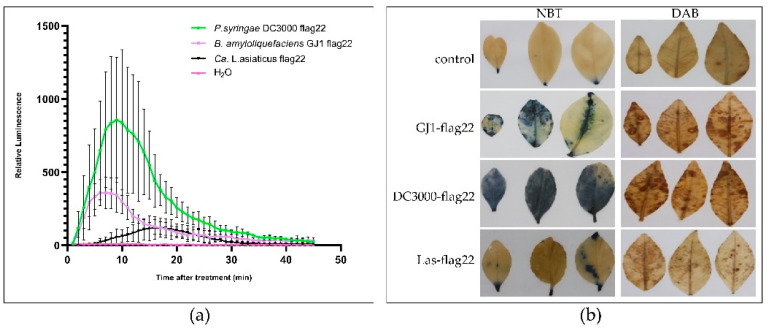
*B. amyloliquefaciens* GJ1 flag22 treatment induced reactive oxygen species (ROS) production. Two-month-old seedling grapefruit (*Citrus paradisi*) trees prepared for test. For the control treatment, the same volume of water was applied. (**a**) The ROS production was measured using the luminol-based assay after treatments with flag22 polypeptide; (**b**) accumulation of O^2−^ and H_2_O_2_ as revealed by histochemical staining with NBT and DAB, respectively.

**Figure 6 plants-10-00261-f006:**
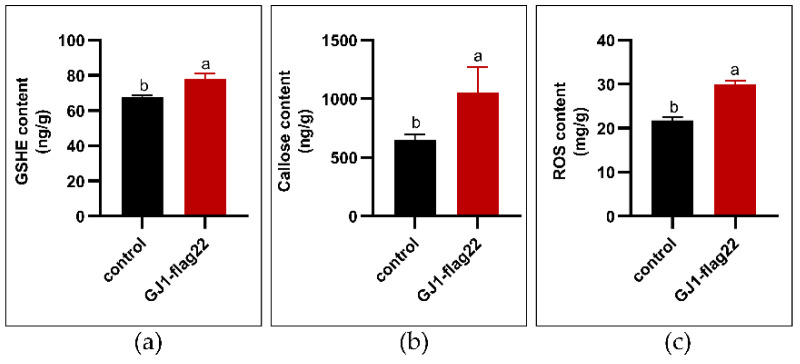
Effect of *B. amyloliquefaciens* GJ1 flag22 treatment on the GSHE (**a**), callose (**b**), and ROS (**c**). Two-month-old seedling grapefruit were treated with *B. amyloliquefaciens* GJ1 flag22 and after 6 h, the mature leaves were harvested for analysis. Bars with different letters (a, b) indicate significant differences (*p* < 0.05) between different treatments.

**Figure 7 plants-10-00261-f007:**
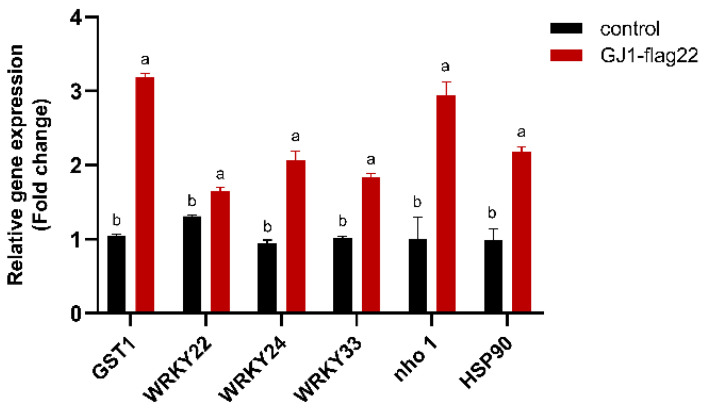
Effect of *B. amyloliquefaciens* GJ1 flag22 treatment on the responses of defense-associated genes. Bars with different letters (a, b) indicate significant differences (*p* < 0.05) between different treatments.

**Figure 8 plants-10-00261-f008:**
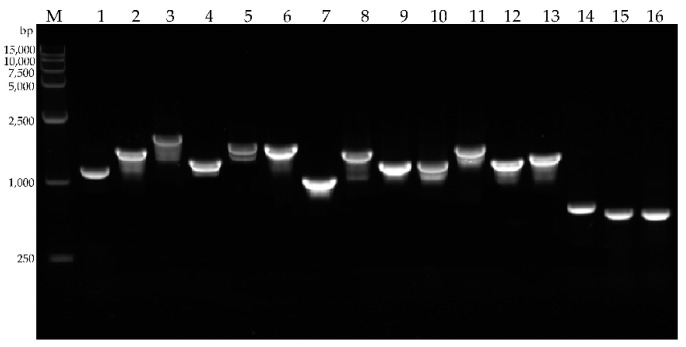
PCR detection of lipopeptide related genes. M: DNA marker, production of 1. *bacA*, 2. *beaS*, 3. *dfnM*, 4. *dfnA*, 5. *dhbA*, 6. *fenA*, 7. *fenE*, 8. *ituF1*, 9. *mlnA*, 10. *mlnI*, 11. *srfAA*, 12. *srfAD*, 13. *ywfG*, 14. *mrsK2*, 15. *mrsK2R2*, and 16. *mrsR2*.

**Figure 9 plants-10-00261-f009:**
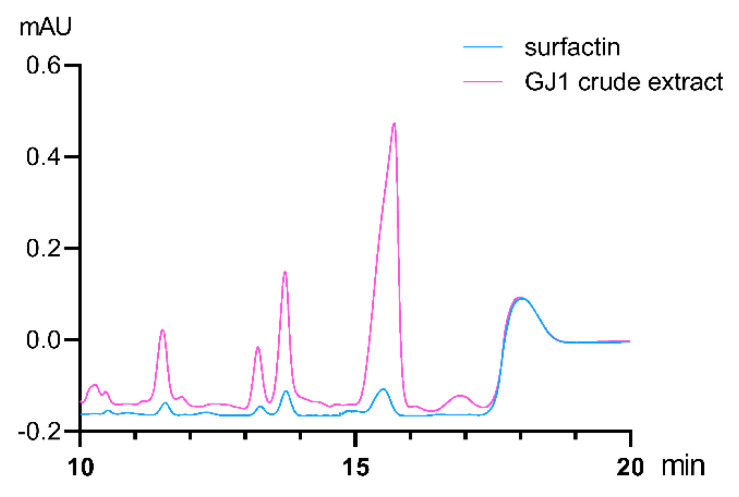
HPLC spectra of surfactin and *B. amyloliquefaciens* GJ1 crude extract.

**Table 1 plants-10-00261-t001:** Quantification results of the three candidate proteins by parallel reaction monitoring (PRM).

Accession	Name	Description	Transcriptome Result	PRM Result
*Cs2g07330*	PsbQ	electron transporter	1.39	1.84
*Cs5g34450*	PsaE	PS1 reaction center subunit III	1.67	2.01
*Cs5g31180*	PsaD	PSI reaction center subunit II	1.44	1.56

**Table 2 plants-10-00261-t002:** Homology analysis of lipopeptides synthetic genes from *B. amyloliquefaciens* GJ1.

Metabolite	Gene	Homology Comparison with *B. amylolyticus* FZB42 (%)
Bacilysin	bacA	98.56
Bacillaene	beaS	97.80
Difficidin	dfnM	98.10
	dfnA	97.73
Bacillibactin	dhbA	96.89
Fengycin	fenA	95.56
	fenE	96.92
IturinA	ItuF1	88.60
Macrolactin	mlnA	96.22
	mlnI	98.39
Surfactin	srfAA	94.25
	srfAD	98.70
Bacilycin	ywfG	98.42
	MrsK2	98.61
	MrsK2R2	99.32
	MrsR2	97.32

**Table 3 plants-10-00261-t003:** Compounds that have been tested against huanglongbing (HLB).

Compound	Application Method	Impact on	Reference
Streptomycin	Greenhouse	Reduction in population density in leaves	[35]
Penicillin G	Field	Reduction of the titer in leaves	[12]
Oxytetracycline	Field	The population density in leaves decreased	[36]
Benzbromarone + tolfenamic acid	Greenhouse	Lower transcription of the CLas 16S rRNA gene was observed	[37]

**Table 4 plants-10-00261-t004:** Primers used for qRT-PCR.

Gene	Accession Number	Primer Sequence	Amplification Length (bp)
*GST1*	LOC102614737	F: GCCCGTTTGTCTCAGTCCAA	59
		R: TGCAAATCGACCAAGGTGAA	
GAPC2	LOC102624117	F: TCTTGCCTGCTTTGAATGGA	80
		R: TGTGAGGTCAACCACTGCGACAT	
*nho 1*	LOC102615775	F: GAACACAGGTGAGAGGTAGTT	91
		R: AGCATAGTTATCGGTGCTTTAG	
*HSP90*	LOC102578032	F: TACCCAATTTCCCTCTGGATTG	97
		R: CCTCAACTTTACCCTCCTCATC	
*WRKY 22*	LOC102622218	F: ACCACAAGTACCACCACAAG	95
		R: CTGGTTTGTTCACGGCTAAATG	
*WRKY 24*	LOC102621617	F: ACCATCACCACCCAACAAA	92
		R: CGGTGCGGAAGATGTAAGAA	
*WRKY 33*	LOC102608921	F: CCGGATTGTCCGATGAAGAAA	98
		R: GATGTAGGCTTGGGATGATTGT	
*Cs4g15270*	LOC102622357	F: CCATGATGGAACTTGAGGGAG	91
		R: GAGTGTAAACGACTGGGGAAG	
*Orange1.1t00226*	LOC102622121	F: GAAAGCCCTTCCGACATACA	118
		R: GTCTGCACTACCACCAAGAA	
*Actin*	LOC102577980	F: CCAAGCAGCATGAAGATCAA	101
		R: ATCTGCTGGAAGGTGCTGAG	

## Data Availability

All data is contained within the article. The datasets analyzed during the current study are available from the corresponding author on reasonable request.

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
