# Peer review of "Antibacterial Potential of Bacillus amyloliquefaciens GJ1 against Citrus Huanglongbing"

_plants, 2021, doi:10.3390/plants10020261_

Round 1

Reviewer 1 Report

This MS is written very poorly and many important details are lacking for readers to understand what, why and when etc was done. With its current version it is not possible to understand or even see the claimed results. The entire MS needs a major revision in terms of details, organization, language and presentation aspects. I have include my specific questions, comments and concerns in attached PDF for authors consideration.

Reviewer 2 Report

I have attached a file with some very minor suggestions/comments. Briefly:

  1. The manuscript would benefit from a read through for English and grammar.
  2. The abstract could do with some work and is a little choppy.
  3. The discussion is overly long and could be shortened considerably.
  4. The discussion and results could be combined?

Author Response

Point 1: Suggest: ... is a serious global concern for the citrus industry.

Response 1: Thanks for your suggestion. We have changed “a serious concern for citrus industries worldwide” to “is a serious global concern for the citrus industry” in line 10.

Point 2: Suggest: ... no established strategy for control of this disease.

Response 2: Thanks for your suggestion. We have changed “strategy against this disease” to “no established strategy for control of this disease” in line 10-11.

Point 3: In this study, compared to the control treatment, under the treatment of B. amyloliquefaciens GJ1, the photosynthetic parameters, chlorophyll content, resistance-associated enzyme content and the expression of defense-realated genes were significantly higher, and the content of starch content and soluble sugar were significantly lower. Reword

Response 3: Thanks for your suggestion. We have reworded this sentence “In this study, two years old citrus infected by Ca. L. asiaticus were treated with B. amyloliquefaciens GJ1 soiution via root irrigation. These plants, after seven irrigation treatments, the results indicated that the photosynthetic parameters, chlorophyll content, resistance-associated enzyme content and the expression of defense-realated genes were significantly higher for the plants treated with the same volume water. The content of starch and soluble sugar were significantly lower, compared to the control treatment” in line 12-17.

Point 4: might

Response 4: Thanks for your suggestion. We have changed “might” to “may” in line 23.

Point 5: Las, Maybe refer to in a different manner or explain in the text.

Response 5: Thanks for your suggestion. We have changed “Las” to “Ca. L. asiaticus” in line 12.

Point 6: in other citrus producing regions

Response 6: Thanks for your suggestion. We have changed “in other citrus producing regions” to “worldwide” in line 38.

Point 7: Suggest: The effects of HLB has been confirmed in 51 of the 140 citrus producing countries.

Response 7: Thanks for your suggestion. We have changed “Up to now, HLB has been confirmed in 51 of the 140 citrus producing countries” to “The effects of HLB have been confirmed in 51 of the 140 citrus producing countries” in line 38-39.

Point 8: Suggest: ... control of the psyllid vectors.

Response 8: Thanks for your suggestion. We have changed “psyllid vectors” to “the psyllid vectors” in line 45.

Point 9: Suggest: Biological control efforts, using microorganisms ....

Response 9: Thanks for your suggestion. We have changed “Biological control is using microorganisms” to “Biological control efforts, using microorganisms” in line 53.

Point 10: Please add P value if it is significant:... were decreased significantly (P...) compared with ...

Response 10: Thanks for your suggestion. We have added the P value in line 80 and 174.

Point 11: by “Candidatus Liberibacter asiaticus”, the starch and sugar in leaves will accumulate, while excessive starch will damage chloroplast function and affect photosynthesis. Reword. Not clear.

Response 11: Thanks for your suggestion. We reword this sentence “Citrus infected by “Candidatus Liberibacter asiaticus” caused Phloem blockage, then the starch and sugar in leaves will accumulate. Excessive starch will damage chloroplast function and finally photosynthesis is repressed” in line 203-205.

Point 12: In this experiment, the effect of B. amyloliquefaciens GJ1 to induce plant resistance against HLB was studied. remove or reword with the following sentence

Response 12: Thanks for your suggestion. We remove this sentence in line 206.

Point 13: showed

Response 13: Thanks for your suggestion. We have changed “showed” to “demonstrate” in line 211.

Point 14: spp

Response 14: Thanks for your suggestion. We have changed “spp” to “spp.” in line 219.

Point 15: become a hot spot.

Response 15: Thanks for your suggestion. We have changed “become a hot spot” to “attracted great attention” in line 240.

Point 16: The drug resistance caused by the abuse of chemical pesticides and the consumer’s demand for sustainable agriculture and nutrition and health are urgently needed to find new prevention methods for pest control. Reword

Response 16: Thanks for your suggestion. We have reworded this sentence “There are urgently needed to find new prevention methods for pest control, because the chemicals used have a negative impact on the environment and human life, and the consumer’s demand for sustainable agriculture” in line 280-283.

Reviewer 3 Report

Nice study but the actual content missing. It would be good to see the comparison of the citrus fruit infected with Liberibacter vs. Liberibacter inoculated with Bacillus amyloliquefaciens.

Line# 12-14: Not clear. Revise the sentence.

Line# 16: …….revealed that Cs2g07330, Cs5g31180, and Cs5g34450 in the photosynthesis pathway were upregulated.

Line# 53: disease problems?

Line# 115: There are several other genes that are involved in defense. Why did you only test WRKY22, 24, and 33?

Round 2

Reviewer 3 Report

The authors have still not responded to my first question. See here,

It would be good to see the comparison on the citrus fruit infected with Liberibacter vs. Liberibacter inoculated with Bacillus amyloliquefaciens.

Author Response

Point 1: It would be good to see the comparison on the citrus fruit infected with Liberibacter vs. Liberibacter inoculated with Bacillus amyloliquefaciens.

Response 1: Thanks for your suggestion. Two years old citrus infected by Ca. L. asiaticus were used for research in this study, some physiological indices (photosynthetic parameters, chlorophyll content, starch and soluble sugar content), and the expression of defense-realated genes were tested after treatment with B. amyloliquefaciens GJ1. We didn't get the fruit, so we do not compare the Liberibacter infected fruit with the Liberibacter infected fruit inoculated with Bacillus amyloliquefaciens GJ1. Next step, we will use six years old citrus infected by Ca. L. asiaticus for the further research, in addition to the above indices, we will also focus the citrus fruit infected with Liberibacter vs. Liberibacter inoculated with Bacillus amyloliquefaciens GJ1.

This manuscript is a resubmission of an earlier submission. The following is a list of the peer review reports and author responses from that submission.